# Pterostilbene Increases LDL Metabolism in HL-1 Cardiomyocytes by Modulating the PCSK9/HNF1α/SREBP2/LDLR Signaling Cascade, Upregulating Epigenetic hsa-miR-335 and hsa-miR-6825, and LDL Receptor Expression

**DOI:** 10.3390/antiox10081280

**Published:** 2021-08-12

**Authors:** Yen-Kuang Lin, Chi-Tai Yeh, Kuang-Tai Kuo, Vijesh Kumar Yadav, Iat-Hang Fong, Nicholas G. Kounis, Patrick Hu, Ming-Yow Hung

**Affiliations:** 1Biostatistics Research Center, Taipei Medical University, Taipei 110, Taiwan; robbinlin@tmu.edu.tw; 2Graduate Institute of Athletics and Coaching Science, National Taiwan Sport University, Taoyuan City 33301, Taiwan; 3Department of Medical Research and Education, Taipei Medical University-Shuang Ho Hospital, New Taipei City 23561, Taiwan; ctyeh@s.tmu.edu.tw (C.-T.Y.); vijeshp2@gmail.com (V.K.Y.); 18149@s.tmu.edu.tw (I.-H.F.); 4Department of Medical Laboratory Science and Biotechnology, Yuanpei University of Medical Technology, Hsinchu 300, Taiwan; 5Department of Surgery, Division of Thoracic Surgery, Shuang Ho Hospital, Taipei Medical University, New Taipei City 23561, Taiwan; doc2738h@gmail.com; 6Department of Surgery, Division of Thoracic Surgery, School of Medicine, College of Medicine, Taipei Medical University, Taipei 110, Taiwan; 7Department of Internal Medicine, Division of Cardiology, University of Patras Medical School, 26221 Patras, Greece; ngkounis@otenet.gr; 8Department of Cardiology, University of California, Riverside, CA 92521, USA; dr.hu.md@gmail.com; 9Department of Cardiology, Riverside Medical Clinic, Riverside, CA 92506, USA; 10Department of Internal Medicine, Division of Cardiology, School of Medicine, College of Medicine, Taipei Medical University, Taipei 110, Taiwan; 11Taipei Heart Institute, Taipei Medical University, Taipei 110, Taiwan; 12Department of Internal Medicine, Division of Cardiology, Shuang Ho Hospital, Taipei Medical University, New Taipei City 23561, Taiwan

**Keywords:** pterostilbene, LDLR, PCSK9, SREBP2, HNF1α, hyperlipidemia, statin

## Abstract

Proprotein convertase subtilisin/kexin type 9 (PCSK9) can promote the degradation of low-density lipoprotein (LDL) receptor (LDLR), leading to hypercholesterolemia and myocardial dysfunction. The intracellular regulatory mechanism by which the natural polyphenol pterostilbene modulates the PCSK9/LDLR signaling pathway in cardiomyocytes has not been evaluated. We conducted Western blotting, flow cytometry, immunofluorescence staining, and mean fluorescence intensity analyses of pterostilbene-treated mouse HL-1 cardiomyocytes. Pterostilbene did not alter cardiomyocyte viability. Compared to the control group, treatment with both 2.5 and 5 μM pterostilbene significantly increased the LDLR protein expression accompanied by increased uptake of LDL. The expression of the mature PCSK9 was significantly suppressed at the protein and mRNA level by the treatment with both 2.5 and 5 μM pterostilbene, respectively, compared to the control. Furthermore, 2.5 and 5 μM pterostilbene treatment resulted in a significant reduction in the protein hepatic nuclear factor 1α (HNF1α)/histone deacetylase 2 (HDAC2) ratio and sterol regulatory element-binding protein-2 (SREBP2)/HDAC2 ratio. The expression of both hypoxia-inducible factor-1 α (HIF1α) and nuclear factor erythroid 2-related factor 2 (Nrf2) at the protein level was also suppressed. Pterostilbene as compared to short hairpin RNA against SREBP2 induced a higher protein expression of LDLR and lower nuclear accumulation of HNF1α and SREBP2. In addition, pterostilbene reduced PCSK9/SREBP2 interaction and mRNA expression by increasing the expression of hsa-miR-335 and hsa-miR-6825, which, in turn, increased LDLR mRNA expression. In cardiomyocytes, pterostilbene dose-dependently decreases and increases the protein and mRNA expression of PCSK9 and LDLR, respectively, by suppressing four transcription factors, HNF1α, SREBP2, HIF1α, and Nrf2, and enhancing the expression of hsa-miR-335 and hsa-miR-6825, which suppress PCSK9/SREBP2 interaction.

## 1. Introduction

Hypercholesterolemia is a complex disease that can directly affect the heart structure and function independently of ischemia [1]. In the general population and patients with or without myocardial ischemia, hypercholesterolemia due to an elevated level of low-density lipoprotein (LDL) cholesterol (LDL-C) has consistently been associated with worse outcomes, including mortality, cardiovascular events, and heart failure [2]. Although LDL, which carries approximately 70% to 75% of total cholesterol [3], causes atherosclerotic coronary artery disease [4], its direct effects on the myocardium and the underlying mechanisms have not been fully explored. On the other hand, serum lipids could accumulate in the heart and alterthe cardiomyocyte mitochondrial function, making the myocardium more vulnerable to damage, which results in cardiac dysfunction [1]. In murine HL-1 cardiomyocytes, the oxidative product of cholesterol can induce oxidative stress and cell death [5]. Moreover, the increase of oxidized LDL (Ox LDL) was correlated with the decrease of the left ventricular ejection fraction [6], leading to the progression of heart failure. Remarkably, lowering serum lipid could reverse early ventricular dysfunction and provide cardioprotection [1], suggesting that elevated plasma level of LDL-C is a modifiable risk factor for myocardial dysfunction. The 2018 cholesterol guidelines from the American College of Cardiology and American Heart Association recommend specific percentage reductions in LDL-C levels as well as the use of LDL-C thresholds for the addition of non-statin therapy to statins in patients at high risk for atherosclerotic cardiovascular events [7]. Therefore, therapy to reduce LDL-C is one of the promising strategies for the prevention of myocardial dysfunction.

The proprotein convertase subtilisin kexin type 9 (PCSK9), identified in 2001 with its gene characterization in 2003 [8], acts as a negative regulator of LDLR by binding to LDLR and therefore increases LDL-C [9], leading to atherosclerotic coronary artery disease [4], with or without endothelial dysfunction [10]. Hypercholesterolemia can further increase PCSK9 [11]. Beyond its role in cholesterol homeostasis, PCSK9 is prevalent in human macrophages [12], smooth muscle cells, endothelial cells [13], and cardiomyocytes, [9] with a local effect that can regulate vascular homeostasis and atherosclerosis [12], suggesting a different role of PCSK9 in the heart. In rat cardiomyocytes, while overexpression of human PCSK9 reduces cell shortening independent of oxidized LDL, silencing of PCSK9 in cardiomyocytes attenuates oxidized LDL-dependent effects on cell shortening [9]. Moreover, oxidized LDL induces PCSK9 release from cardiomyocytes into the supernatant [9]. Hence, PCSK9 acts in a direct, autocrine way on cardiomyocytes and impairs their function [9]. A novel secretome analysis of primary cardiomyocytes led to the identification of PCSK6 as being vitally involved in adverse cardiac remodelling under myocardial ischemia and infarction in mice and humans [14]. The data from genetic, epidemiological, and pharmacological studies have led to the conclusion that inhibiting PCSK9 reduces cardiovascular events [15]. Because PCSK9 genetic gain-of-function mutations are associated with hypercholesterolemia, the pharmacological inhibition of PCSK9 is considered a promising route of intervention for preventing cardiovascular diseases [16]. However, the intracellular mechanism in cardiomyocytes that affects PCSK9 expression remains undetermined.

To reduce PCSK9 levels or its LDLR-binding, either the monoclonal antibodies or antisense oligonucleotides strategies can be applied [17]. Two monoclonal antibodies, alirocumab and evolocumab, are currently approved for the treatment of hypercholesterolemia and achieved significant reductions in LDL levels as well as levels of apolipoprotein B and lipoprotein(a) [18]. On the other hand, a third approach to inhibit PCSK9 may include the pharmacological development and validation of nature-derived orally absorbed compounds that have anti-PCSK9 activity properties, among which polyphenols, such as resveratrol, have been reported to exert many cardiovascular benefits through multiple mechanisms of action, including LDL-C lowering activity, in several epidemiological studies and clinical trials [15]. Regarding the six mechanisms of PCSK9 inhibition, while most nature-derived compounds, including resveratrol, inhibit transcription factors of PCSK9, for example, sterol regulatory element-binding protein-2 (SREBP2) and hepatocyte nuclear factor 1α (HNF1α), some utilize the other three mechanisms that inhibit PCSK9/LDLR interaction, such as resveratrol, the maturation of PCSK9 in the endoplasmic reticulum, or the secretion of PCSK9 [15]; however, to date, there is still lacking evidence to support that nature-derived compounds can effectively affect PCSK9 at the translational level and by epigenetic mechanisms [15]. Among polyphenols, pterostilbene, a plant-derived stilbenoid antioxidant and analogue of resveratrol, differs from resveratrol through exhibiting induced lipophilic and oral bioavailability (80% compared to 20% in resveratrol) because of the presence of the two methoxy groups [19], which makes it potentially advantageous as a therapeutic agent. While resveratrol transcriptionally reduces PCSK9 promoter activity and inhibits the interaction of PCSK9 with LDLR [15], whether pterostilbene similarly inhibits PCSK9 remains unknown. Although pterostilbene has been demonstrated to exert cardioprotection against myocardial ischemia via attenuating inflammation [20], pterostilbene mechanisms vary in each disease system. Thus, the treatment of pterostilbene may upregulate or downregulate specific pathways based on the nature of the disease [21]. Collectively, it remains unknown whether pterostilbene could mediate the PCSK9/HNF1α/SREBP2/LDLR signaling pathway in cardiomyocytes by more than one known and unknown mechanism of action of PCSK9 inhibition.

SREBP2 has been reported to preferentially activate genes involved in cholesterol biosynthesis and metabolism, such as the activation of 3-hydroxy-3-methylglutaryl coenzyme A synthase and reductase [22], resulting in intracellular cholesterol accumulation. The SREBP pairing with HNF1α, whose binding sites reside at 28 bp upstream from the SREBP2 binding element as a predominant sequence motif for PCSK9 transcription, to control PCSK9 expression was firstly described in 2009 [23]. However, PCSK9 expression can be increased through SREBP2-independent pathways in inflammatory states [24]. Furthermore, the expression of PCSK9 can also be altered by oncologic pathways with the utilization of protein kinases, such as Akt [25]. On the other hand, other transcription factors may be involved in lipid accumulation under hypoxia and inflammation [26]. Hypoxia-inducible factor-1 α (HIF1α) promotes intracellular cholesterol accumulation through inducing peroxisome proliferator-activated receptor γ [27]. Nuclear factor erythroid 2-related factor 2 (Nfr2) activation contributes to the advanced stage of atherosclerotic plaque formation but not in earlier lesions [28]. Therefore, finding a new regulatory network modulating PCSK9/LDLR transcription is essential for the development of selective inhibitors of PCSK9 to improve the treatment efficacy via extending the up-regulation of LDLR. As a result, the purposes of our study in in vitro HL-1 cardiomyocytes were to investigate (1) the toxicity of pterostilbene on cells; (2) the effects of pterostilbene on PCSK9 expression, LDLR expression, and LDL-C metabolism; (3) whether more than one intracellular mechanism was involved in regulating pterostilbene-mediated PCSK9 expression; and (4) the influence of pterostilbene on epigenetic non-coding microRNAs (miRNAs) expression associated with the regulation of PCSK9.

## 2. Materials and Methods

### 2.1. Reagents and Drugs

Pterostilbene (>97% high-performance liquid chromatography (HPLC)-pure) was purchased from Sigma-Aldrich (St. Louis, MO, USA). Pterostilbene was dissolved in dimethyl sulfoxide (DMSO) and further diluted in a sterile culture medium before use. DMSO, Sulforhodamine B dye, Dulbecco’s minimum essential medium (DMEM), trypsin/EDTA, phosphate-buffered saline (PBS), TRIS base, and acetic acid were also procured from Sigma-Aldrich (St. Louis, MO, USA). Alexa Fluor 488 and Alexa Fluor 647 donkey anti-rabbit IgG antibodies were purchased from Invitrogen (Grand Island, NY, USA). A TRIzol Plus RNA isolation and purification kit was obtained from Life Technologies (Invitrogen, Rockville, MD, USA), and real-time polymerase chain reaction (rt-PCR) primers, dNTP, reverse transcriptase, and Taq polymerase were obtained from Qiagen (Valencia, CA, USA).

### 2.2. Cells and Cell Culture

The mouse HL-1 cardiomyocyte cell line (SCC065) was obtained from Sigma-Aldrich (St. Louis, MO, USA) and cultured in the DMEM media supplemented with 10% fetal bovine serum (FBS) and 1% penicillin-streptomycin (Thermo Fisher Scientific, Rockford, IL, USA) for 24 h, and then it was transferred to DMEM medium supplemented with 10% lipoprotein deficient serum (Sigma-Aldrich); this mixture was maintained overnight at 37 °C in a humidified 5% CO_2_ incubator. After reaching ≥95%, the confluency cells were passaged. For treatment, the cells were incubated in a medium containing a vehicle (0.1% DMSO) or pterostilbene in a humidified incubator at 37 °C and 5% CO_2_ for 24 h.

### 2.3. Western Blot Analysis

After trypsinization and harvesting of vehicle- or pterostilbene-treated HL-1 cardiomyocytes, the cells were lysed. NE-PER nuclear and cytoplasmic extraction reagents (#78835; Thermo Fisher Scientific, Waltham, MA, USA) was used to isolate proteins and protein lysates heated for 5 min, followed by Western blotting. The blotting signals were blocked with 5% skim milk in Tris-buffered saline (TBS) with Tween 20 for 1 h, before overnight incubation at 4 °C with respective primary antibodies against PCSK9, LDLR, HIF1α, Nrf2, Akt, p-Akt, HNF-1α, SREBP2, histone deacetylase 2 (HDAC2), and β-actin (Appendix A). Next, the polyvinylidene difluoride membranes were washed twice with TBS saline with Tween 20, incubated with a secondary antibody kept at room temperature for 1 h, and again washed with TBS buffer with Tween 20. Protein bands were visualized using the enhanced chemiluminescence Western blot reagents and the iBright Western Blot Imaging System (Thermo Fisher Scientific, Waltham, MA, USA).

### 2.4. Sulforhodamine B Cell Viability Assay

HL-1 cardiomyocytes were seeded at 2.5 × 10^3^ cells/well in 96-well microtiter plates in a complete culture medium and incubated in a humidified incubator with at 37 °C and 5% CO_2_ for 24 h. After pterostilbene treatment, cell viability was measured using the sulforhodamine B (SRB) assay, as previously described [29]. DMSO-treated cells were used as a control. Assays were performed thrice in triplicates and the absorbance was measured at 510 nm by a Geminin XPS microplate reader (Molecular Devices, San Jose, CA, USA).

### 2.5. shSREBP2 (Short Hairpin RNA against SREBP2) Knockdown Procedure

The loss of function of SREBP2 in the HL-1 cardiomyocyte cells was assayed by using commercially available systems. SREBP2 gene-knockdown shRNA sets (expression Arrest GIPZ Lentiviral shRNA) were procured from Thermo Fisher Scientific (Bartlesville, OK, USA). SREBP2 clones were used to silence SREBP2 expression, and a non-silencing verified negative control acting as the control. The lentiviral particles were produced for loss-of-function studies were conducted as per the manufacturer’s protocols.

### 2.6. RT-qPCR of hsa-miR-335 and hsa-miR-6825 Assay

For miR assay, the 5 × 10^4^ HL-1 cardiomyocyte cells were plated into 24-well plates 24 h before going for transfection. As per the manufacturer’s instruction, the mi-181d (control), Syn miR, mimic or inhibitor was transfected using the Hiperfect Transfection Reagent (#301705, QIAGEN Inc., Germantown, MD, USA). All the miScript inhibitor, negative control, Syn-hsa-miR-335 and hsa-miR-6825 miScript miRNA mimic, anti- hsa-miR-335 and hsa-miR-6825 miScript miRNA inhibitor, and hsa-miR-335 and hsa-miR-6825 miScript primer assay were purchased from QIAGEN Inc., Germantown, MD, USA. Shortly after the adding Hiperfect transfection reagent (3 µL) to FBS-free DMEM (100 µL) containing hsa-miR-335 and hsa-miR-6825 mimic or the control (negative) to a 50 nmol/L final concentration and the wells were adjusted to 600 µL with 10% FBS supplemented culture media. To evaluated the transfection efficiency, green fluorescent protein (GFP) expression was observed under the fluorescence microscope. We also determined the hsa-miR-335 and hsa-miR-6825 expression in transfected HL-1 cardiomyocyte cells by real-time PCR.

### 2.7. Real-Time Reverse Transcriptase PCR

We isolated and purified the total RNA by using the TRIzol Plus RNA purification kit (Life Technologies) as per the vendor’s instruction. A master mix of cDNA, gene primers, and the Maxima SYBR Green/ROX qPCR Master Mix 2X (#K0221; Thermo Fisher Scientific) were prepared to perform the real-time qPCR assay, and PCR amplification was carried out in an iCycler iQ detection system (BioRad, Hercules, CA, USA) following the cycling conditions: 58 °C for 2 min, 94 °C for 4 min, and 40 cycles of 94 °C for 1 min, 58 °C for 1 min, and 72 °C for 1 min. β-actin-normalized mRNA expression was quantified using ∆∆Ct method.

### 2.8. Flow Cytometry Detection of Cell-Surface LDLR

Flow cytometry assay as per the process described previously [30] was carried out to detect the cell-surface LDLR expression. Briefly, cells were seeded and treated with pterostilbene or the DMSO control for 24 h. After treatment cells were detached, washed with 1 × PBS, and incubated at room temperature with 1 × PBS/5% bovine serum albumin-blocking reagent buffer for 30 min. The cells were incubated with anti-LDLR antibodies for 1 h at 37 °C and washed with PBS, and finally incubated with Alexa Fluor 488-conjugated goat anti-rabbit IgG (Thermo Fisher Scientific) antibodies at room temperature for 30 min, followed by LDLR expression detection through flow cytometry by using the FACScan system (BD Biosciences, San Jose, CA, USA). The flow cytometry data were analyzed using the BD CellQuest Pro-Software v6.0 (BD Biosciences). The LDLR-cell surface expression level data was expressed and plotted in the relative percentage of the geometric mean of fluorescence intensity.

### 2.9. Intracellular LDL Uptake Evaluation

The intracellular level of LDL uptake was assayed as per the method described earlier [30] by using the Image-iT Low-Density Lipoprotein Uptake Kit, BODIPY FL (#I34359; Thermo Fisher Scientific). Briefly, after the treatment with pterostilbene or the vehicle for 24 h, removal of the treatment medium, and replacement with serum-free DMEM, the HL-1 cardiomyocytes were treated with BODIPY FL LDL (10 μg/mL) (Thermo Fisher Scientific) for 24 h, at 37 °C. After the treatment, the cells were collected, washed with PBA and resuspended in 1 × PBS and then analyzed with flow cytometry, with the data acquired and stored from 10,000 cells (counts). The LDL uptake was measured and expressed in a relative percentage of the geometric mean of fluorescence intensity.

### 2.10. Immunocytochemistry Assessment of Intracellular LDL Density

For the assessment of LDL reactivity in the HL-1 cardiomyocytes, we used the LDL Uptake Assay Kit (cell-based) (ab133127; Laizee Biotech, Shanghai, China), following the manufacturer’s instructions. Briefly, after the removal of the culture medium from treated cells, the cells were incubated in LDL-DyLight 550 solution for 4 h, placed back into the culture medium, and analyzed for LDL uptake under fluorescence microscopy. Next, after the 1 × PBS wash cells were fixed for 10 min, then again washed with 1 × PBS thrice for 5 min intervals. After wash cells were incubated with an assay blocking solution for 30 min, and then incubated with antibody against anti-LDLR for 1 h, washed with PBS again to remove the unbound antibodies. Furthermore, the primary-LDLR antibody labelled cells were incubated with Dylight 488 secondary antibody for 1 h, washed, and then analyzed under a fluorescence microscope.

### 2.11. Plasmid Transfection and PCSK9 Promoter/Luciferase Reported Assay

Transfection of reported plasmids and the establishment of stable PCSK9 promoter –cells, and the luciferase report assay were conducted strictly following already established protocols [31] and according to the reagent manufacturers’ instructions. Transfected cells were then treated with DMSO- or pterostilbene for 24 h. Luciferase activity was evaluated using the Luciferase Reporter Assay System (#E1500; Promega) and all the luminescence was Renilla luciferase activity-normalized.

### 2.12. Statistical Analysis

All the experiments were performed at least three times in triplicates, and the values were measured as a mean ± standard deviation. A one-way ANOVA with Dunnett’s post hoc test was performed using GraphPad Prism v5 for Windows (GraphPad Software, La Jolla, CA, USA) and the *p* < 0.05 was reflected statistically significant. 

## 3. Results

### 3.1. Pterostilbene Exhibits No Apparent Cytotoxicity but Significantly Enhances LDLR Expression

To determine the effect of pterostilbene on the cell viability of human cardiomyocytes, HL-1 cardiomyocytes, which have been shown to proliferate in culture media without losing their cardiac-specific phenotype, were treated with an increasing concentration of pterostilbene up to 5 μM. Pterostilbene did not affect the viability compared to the control cells (Figure 1A). The Western blot analysis demonstrated, as compared to the control group, the LDLR protein expression was significantly and dose-dependently increased 1.31-fold (*p* < 0.05) and 1.60-fold (*p* < 0.01) by treatment with 2.5 and 5 μM pterostilbene, respectively (Figure 1B). This finding was replicated with immunocytochemistry staining: a 1.24-fold (*p* < 0.05) and 1.33-fold (*p* < 0.01) upregulation of cell-surface LDLR protein expression was observed after treatment with 2.5 and 5 μM pterostilbene, respectively (Figure 1C). 

### 3.2. Pterostilbene Enhances the Cell-Surface LDLR Expression and the Uptake of LDL

Flow cytometry analysis demonstrated that pterostilbene enhanced cell-surface LDLR expression (Figure 2A,B), which was associated with an increased intracellular accumulation of LDL particles in a dose-dependent manner (Figure 2C). Furthermore, we investigated whether increased cell-surface LDLRs were associated with enhanced activities. As shown in Figure 2D, 2.5 and 5 μM pterostilbene significantly (*p* < 0.01, and *p* < 0.001) increased the uptake of LDL approximately 1.4-fold and 1.65-fold, respectively, compared to the vehicle control group.

### 3.3. Pterostilbene Suppresses PCSK9/HNF1α/SREBP2 Signaling

To estimate the effect of pterostilbene on PCSK9 and the associated signaling pathways, Western blot analyses were performed and demonstrated that pterostilbene markedly and dose-dependently suppressed the expression of not only the proprotein (p), mature (m), and secreted (s) forms of PCSK9 (Figure 3A), but also the signaling molecules, such as HNF1α and SREBP2, whereas no effect on HDAC2 expression was observed (Figure 3D). In comparison to the control group, the treatment with 2.5 and 5 µM of pterostilbene significantly and dose-dependently suppressed the expression of mature PCSK9 by 30% (*p* < 0.05) and 64% (*p* < 0.01), respectively, at the protein level (Figure 3B), and 27% (*p* < 0.01) and 49% (*p* < 0.01), respectively, at the mRNA level (Figure 3C). As shown in Figure 3E,F, 2.5 and 5 μM pterostilbene resulted in 19.4% (*p* < 0.05) and 60.7% (*p* < 0.01) reduction, respectively, in the HNF1α/HDAC2 ratio (Figure 3E) and 24.1% (*p* < 0.05) and 32.6% (*p* < 0.01) decrease, respectively, in the SREBP2/HDAC2 ratio (Figure 3F). Collectively, an exposure-response relationship exists between pterostilbene and the PCSK9/HNF1α/SREBP2/LDLR signaling pathway.

### 3.4. Pterostilbene Suppresses the PCSK9 Promoter Activity and Hyperlipidemia-Associated Transcription Factors

To investigate whether pterostilbene regulated PCSK9 promoter activity as previously described [31], we amplified the human PCSK9 gene promoter by polymerase chain reaction (PCR), with HL-1 genomic DNA serving as our template, to generate pGL3-Basic vector clones (Promega, Madison, WI, USA) at the Xho I and Hind III sites, resulting in the pGL3-PCSK9-P plasmid, which was used as a template for PCR amplification of the PCSK9 gene promoter fragments (P1–P6) (Figure 4A). Using the constructed luciferase reporter plasmid containing fragments of the 5′ flanking 1455-bp promoter region (P1–P6) of PCSK9, the initial post-transfection amplification of PCSK9 promoter-luciferase activity (10-fold, *p* < 0.01) in HL-1 cardiomyocytes was significantly and dose-dependently suppressed after treatment with 2.5 (5.04-fold, 5.04-fold, *p* < 0.01) and 5 μM (2.85-fold, *p* < 0.01) pterostilbene as shown in Figure 4A. In similar assays, we demonstrated that the marked increase in luciferase activities in the constructs P1–P6 (*p* < 0.01) after promoter transfection into the HL-1 cardiomyocytes was significantly reduced after treatment with 5 μM pterostilbene (*p* < 0.01); however, the reduction in construct P6 was not significant (Figure 4B). Sequence analysis also indicated that the PCSK9 promoter contains binding sites for HNF-1α and SREBP2 in the −411 to −335 bp region (Figure 4C). Moreover, as shown in Figure 4D, treatment with 0, 2.5, and 5 μM pterostilbene for 48 h significantly decreased the activities of hyperlipidemia-associated transcription factors SREBP2, HNF1α, HIF1α, and Nrf2 dose-dependently as compared with non-treated control cells.

### 3.5. Pterostilbene as Compared to shSREBP2 Induced Higher Protein Expression of LDLR and Lower Nuclear Accumulation of HNF-1α and SREBP2 

We determined the molecular mechanisms underlying the pterostilbene-mediated suppression of PCSK9 protein expression by evaluating the effects on the compartmentalized expression of selected molecular factors, with or without shSREBP2. In Western blot results, pterostilbene compared with shSREBP2 decreased PCSK9 and increased LDLR expression to a greater extent. As compared to the control group, the combination treatment of shSREBP2/pterostilbene achieved significantly greater suppression of cytoplasmic expression of PCSK9, p-AKT, AKT, secreted expression of PCSK9, and nuclear expression of HNF1α and SREBP2, while the cytoplasmic LDLR expression was significantly enhanced. The combination treatment effects or pterostilbene alone effect were more potent than those of treatment with shSREBP2 alone (Figure 5A,B).

### 3.6. Pterostilbene Exerts Epigenetic Regulation of PCSK9, SREBP2, and LDLR mRNA Expression through hsa-miR-335 and hsa-miR-6825, Which Mediate the PCSK9/SREBP2 Interaction

To investigate the role of pterostilbene in the epigenetic control of LDL metabolism, bioinformatics analyses for miRNA-target predictions to identify which miRNAs regulate PCSK9/SREBP2 interaction network yielded several potential candidates (Figure 6A), among which hsa-miR-335-5p and hsa-miR-6825-5p were the most likely ones (Figure 6B). We subsequently transfected HL-1 cardiomyocytes with mimics or inhibitors of hsa-miR-335-5p and hsa-miR-6825-5p to validate the predicted interactions. Furthermore, we determined the effect of pterostilbene on these epigenetic modulators of PCSK9/SREBP2/LDLR signaling. We found that treatment with pterostilbene significantly and dose-dependently upregulated the expression of hsa-miR-335-5p (2.5 μM: 1.74-fold, *p* < 0.05; 5 μM: 2.32-fold, *p* < 0.01) and hsa-miR-6825-5p (2.5 μM: 2.28-fold, *p* < 0.001; 5 μM: 4.75-fold, *p* < 0.001) (Figure 6C), indicating that the lipid-lowering effects may be mediated by the upregulated expression of hsa-miR-335-5p and hsa-miR-6825-5p (Figure 6C). On the other hand, compared with the control cells, PCSK9 and SREBP2 mRNA expression was significantly upregulated following the inhibition of miR-335-5p (1.65-fold, *p* < 0.01 and 1.41-fold, *p* < 0.05, respectively) or miR-6825-5p (2.28-fold and 1.83-fold, respectively, both *p* < 0.001), whereas LDLR mRNA expression was markedly suppressed after inhibiting miR-335-5p (0.43-fold, *p* < 0.05) or miR-6825-5p (0.21-fold, *p* < 0.01) (Figure 6D). Conversely, transfection of miRNA mimics decreased PCSK9 and SREBP2 mRNA expression but increased LDLR mRNA expression (Figure 6D).

## 4. Discussion

We found that while pterostilbene did not alter cardiomyocyte viability, it dose-dependently enhanced and suppressed the protein expression of LDLR and PCSK9 in HL-1 cardiomyocytes, respectively. Furthermore, pterostilbene increased LDLR activity and the subsequent LDL uptake. Pterostilbene also suppressed the expression of four transcription factors that control cholesterol homeostasis, including HNF1α, SREBP2, HIF1α, and Nrf2. Pterostilbene as compared to shSREBP2 induced higher protein expression of LDLR and lower nuclear accumulation of HNF-1α and SREBP2. In addition, pterostilbene reduced PCSK9/SREBP2 interaction and mRNA expression by increasing the expression of hsa-miR-335 and hsa-miR-6825, which in turn increased LDLR mRNA expression. These novel inhibitory pathways of pterostilbene, related to transcriptional and epigenetic PCSK9 antagonism, and LDLR upregulation may lead to new approaches to cardiovascular prevention.

Similar to previous human and animal models from preclinical and clinical trials, pterostilbene had no significant toxic effects on cardiomyocytes in our study [32,33]. A randomized, double-blind, placebo-controlled study of healthy people demonstrated the safety for human use at dosages up to 250 mg per day after receiving pterostilbene for 6–8 weeks [34]. While most studies using polyphenols have been performed in human cancer cell lines, some polyphenols exhibit cytotoxicity in a cell type-selective manner [35]. Although toxicity of stilbenes toward normal cell lines is not known [36], cytotoxic effects in non-cancer cell lines have been shown after pterostilbene exposure. Notably, few studies have been performed on normal or non-cancer cell lines and the results are inconsistent depending on the stilbene used, cell lines tested, assay performed, and exposure circumstances [37]. Hence, it is important to evaluate the effects of these stilbenes in normal cell lines to assure their safety before medicinal purposes in humans can occur. In general, a pterostilbene-induced decrease in cell proliferation has been observed in a time- and dose-dependent mannerat low concentrations such as 40 µM [38]. In this regard, our findings are relevant, because normal or non-cancerous cells are more sensitive, and the results may support human extrapolation [39].

In models of hyperlipidemia, pterostilbene increased expression of the lipid-lowering peroxisome proliferator-activated receptor γ [21], which in turn increased the expression of both PCSK9 and LDLR via SREBP2 processing [40]. Additionally, in vivo and in vitro studies showed that pterostilbene acted as a more effective LDL-C lowering peroxisome proliferator-activated receptor α agonist than resveratrol. [41]. We demonstrated that pterostilbene up-regulated LDLR protein and mRNA expression and the subsequent LDL uptake, while down-regulating PCSK9 protein and mRNA expression. Furthermore, a coordinate reduction of HNF1α and SREBP2 by pterostilbene led to a strong suppression of PCSK9 transcription through these two critical regulatory transcription factors. Because of the coexistence of an SREBP2 binding element in the LDLR and PCSK9 promoters, statin treatment leads to increased transcription of both LDLR and PCSK9 [42], which may reduce the treatment efficacy. More importantly, in the liver of dyslipidemic hamsters, rosuvastatin treatment increased HNF1α and SREBP2 expression, which is likely the underlying mechanism accounting for the higher induction of PCSK9 than LDLR because of the utilization of two transcription factors (HNF1α and SREBP2) binding to the PCSK9 promoter versus one (SREBP2) binding to the LDLR promoter [42]. Furthermore, when intracellular cholesterol concentrations are low (as after statin treatments), SREBP2 is activated to promote PCSK9 and LDLR transcription [22]. This intrinsic regulatory loop leads to statin resistance in further lowering plasma LDL-C [42]. On the other hand, normoxic cardiomyocytes use long-chain fatty acids as the principal substrates for adenosine triphosphate energy production, while ischemic cardiomyocytes adapt to consume less oxygen by shifting adenosine triphosphate production from mitochondrial fatty acid β-oxidation to glycolysis facilitated by HIF1α [26]. As a result, HIF1α mediates hypoxia-induced downregulation of fatty acid metabolism by interacting with the peroxisome proliferator-activated receptor α, leading to intracellular lipid accumulation [26]. Another transcription factor Nrf2 promotes atherosclerosis by raising LDL levels in ApoE knockout mice [43]. In our study, in addition to downregulating the expression of HNF1α and SREBP2, pterostilbene also decreased the protein expression of HIF1α and Nrf2, which has not been previously investigated, implicating the role of pterostilbene as a potentially promising treatment strategy against hypercholesterolemia. 

In the modulation of the immune system, nutrition might play an important role through altering the gene expression, whereas only a few studies have shown that natural phenolic compounds affect the treatment of diseases other than cancer, as well as the connection between phenolic compounds and epigenetic changes [44]. A study conducted by Jayakumar T et al., 2021 also demonstrated pterostilbene’s antioxidant defense and anti-inflammatory role via NF-κB/ERK signaling pathways [45]. Epigenetic modulation using oligonucleotides to regulate disease gene expression through a variety of processes, such as RNAi, splicing modulation, target degradation by RNase H-mediated cleavage, non-coding RNA inhibition, programmed gene editing, and gene activation, has opened new opportunities for therapy development [46]. Inclisiran is a double-stranded small interfering RNA designed to target PCSK9 mRNA and subsequently inhibit PCSK9 synthesis and secretion of both free and lipoprotein-bound forms [47]. It has the potential benefit of much lower dose frequency than treatment with PCSK9 monoclonal antibodies. A phase 2 randomized trial, in which inclisiran was compared with a placebo, showed dose-dependent reductions in PCSK9 and LDL cholesterol levels [48]. ORION-4 (A Randomized Trial Assessing the Effects of Inclisiran on Clinical Outcomes among People with Cardiovascular Disease; NCT03705234) is a large, ongoing, phase 3 trial investigating whether inclisiran can reduce major cardiovascular events in adults with established cardiovascular disease [7]. In our study, pterostilbene not only suppressed the critical elements in the proximal promoter for PCSK9 transcription, including binding motifs for the HNF1α and SREBP2, but also prevented nuclear translocation of HNF1α and SREBP2. In addition, pterostilbene induced lower PCSK9 and higher LDLR expression than shSREBP2. The shRNA-mediated PCSK9 knockdown is similar to the findings of an in vitro study that Ad-shHNF1α infection in HepG2 hepatocellular carcinoma cells markedly suppressed PCSK9 mRNA expression and upregulated LDLR protein and that injecting normal diet-fed mice with Ad-shHNF1α significantly suppressed the liver PCSK9 mRNA expression, serum PCSK9, and circulating LDL-C levels while upregulating hepatic LDLR protein expression [49]. On the other hand, we demonstrated for the first time that pterostilbene can suppress PCSK9 epigenetically through hsa-miR-335 and hsa-miR-6825 mediated downregulation of SREBP2, leading to increased mRNA expression of LDLR. The reduced gene expression following transfection of miRNA mimics suggests that the miRNA under study is associated with the regulation of that gene. A recent study has shown that pterostilbene protects against lipid accumulation by modulating the miR-34a/Sirt1/SREBP1 pathway [50], which corroborates the importance of miRNAs in modulating hyperlipidemia. Comparative studies of gene silencing and monoclonal antibody-mediated PCSK9 inhibition will delineate the clinical importance of the intracellular effects of PCSK9, as both intra- and extracellular effects of PCSK9 will be influenced by miRNA-mediated PCSK9 inhibition, while extracellular effects of PCSK9 will be influenced only by monoclonal antibodies. 

Nevertheless, one of the limitations of this present work is that we cannot state whether the pterostilbene metabolites are more efficient than the parent compound or not. For, further, elucidate the molecular mechanisms underlying the role of pterostilbene metabolite, our ongoing study tries to analyze several metabolic pathways involved in triglyceride accumulation, coming from their effects on *de novo* lipogenesis.

## 5. Conclusions

In cardiomyocytes, pterostilbene dose-dependently decreases and increases the protein and mRNA expression of PCSK9 and LDLR, respectively. The cardioprotective benefits of pterostilbene in LDL-C lowering were mediated through the PCSK9/HNF1α/SREBP2/LDLR signaling pathway by mechanisms that involve inhibiting HNF1α/SREBP2/HIF1α/Nrf2 transcription factors, decreasing the secretion of PCSK9, upregulating epigenetic control of miR-335 and miR-6825, and increasing LDLR expression (Graphical Abstract), suggesting pterostilbene’s future clinical use in association with other drugs targeting PCSK9. In clinicalpractice, people with hyperlipidemia in the early stages may have damaged cardiac structure and function without signs or symptoms.As a result, the progress in this field holds great promise of guiding cardioprotection to benefit patients with early-stage hyperlipidemia.

## Figures and Tables

**Figure 1 antioxidants-10-01280-f001:**
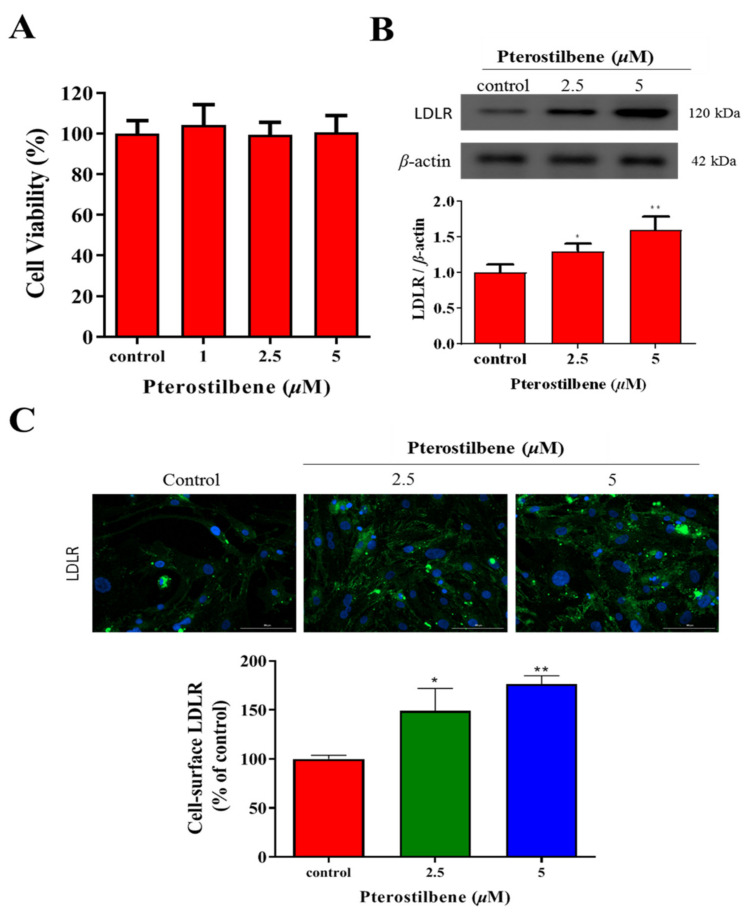
Pterostilbene exhibits no apparent cytotoxicity but significantly enhances LDLR expression in HL-1 cells. (**A**) Histogram showing the effect of pterostilbene on cell viability. (**B**) Representative Western blots image and histograms of the effects of pterostilbene on LDLR protein expression. β-actin was the loading control. (**C**) Immunocytochemistry images showing the enhancing effects of pterostilbene on cell-surface LDLR expression. * *p* < 0.05, ** *p* < 0.01.

**Figure 2 antioxidants-10-01280-f002:**
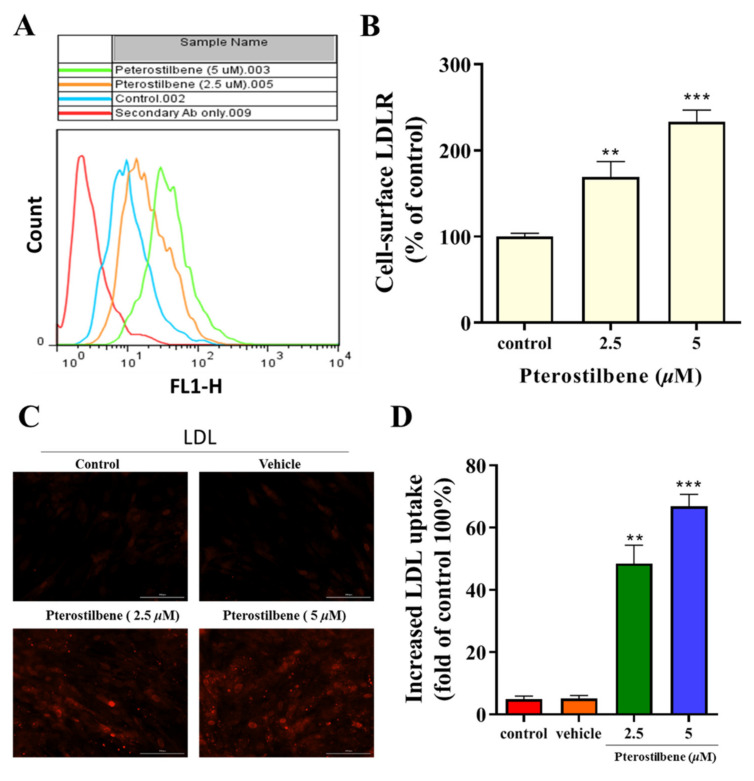
Pterostilbene enhances the LDLR cell surface expression and the uptake of LDL in HL-1 cells. (**A**) Flow cytometry analysis histograms showing a clear distinction between the effect of isotype control secondary antibody, vehicle, and pterostilbene on LDL uptake activity. (**B**) Histograms showing the enhancing effect of pterostilbene on cell-surface LDLR expression. (**C**) Immunocytochemical staining showing the effect of pterostilbene on the intracellular accumulation of LDL particles. (**D**) Effect of pterostilbene on LDL uptake. HL-1 cells were incubated with the fluorescently labelled LDL particles in the presence or absence of the pterostilbene (0, 2.5, and 5 µM), after the treatment the fluorescence intensity was measured. ** *p* < 0.01 and *** *p* < 0.001 as compared to the control.

**Figure 3 antioxidants-10-01280-f003:**
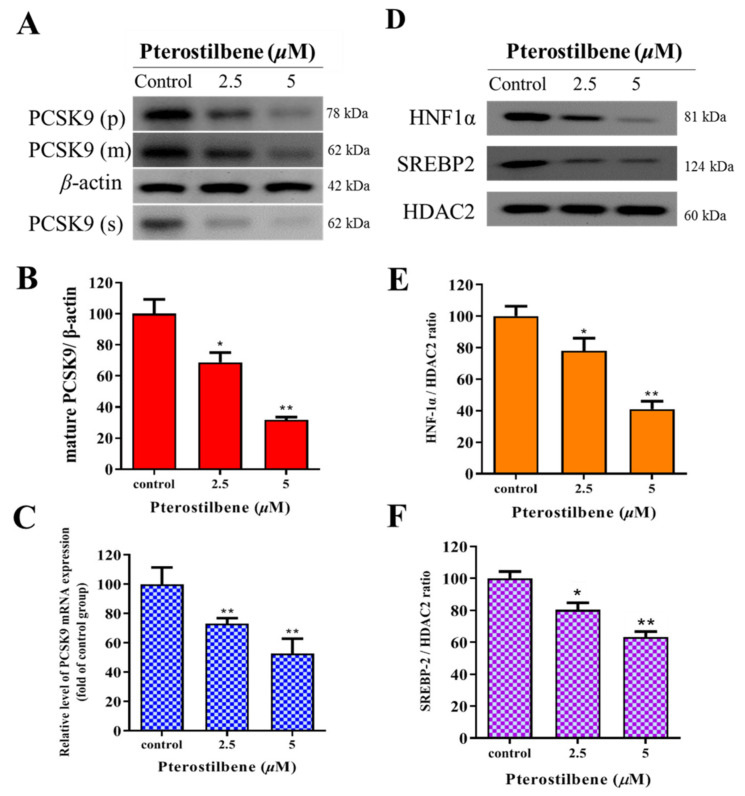
Pterostilbene suppresses PCSK9/HNF1α/SREBP2 signaling in HL-1 cells. (**A**) Representative Western blot images showing the dose-dependent effect of pterostilbene on PCSK9(p), PCSK9(m), and PCSK9(s). β-actin was the loading control. Histograms showing how pterostilbene affects the expression of (**B**) mature PCSK9 protein and (**C**) PCSK9 mRNA. (**D**) Representative Western blot images showing the dose-dependent effect of pterostilbene on HNF-1α, SREBP2, and HDAC2. Histograms of the dose-dependent effect of pterostilbene on (**E**) HNF-1α/HDAC2 ratio and (**F**) SREBP2/HDAC2 ratio. PCSK9(p), proprotein PCSK9; PCSK9(m), mature PCSK9; PCSK9(s), secreted PCSK9. * *p* < 0.05, ** *p* < 0.01.

**Figure 4 antioxidants-10-01280-f004:**
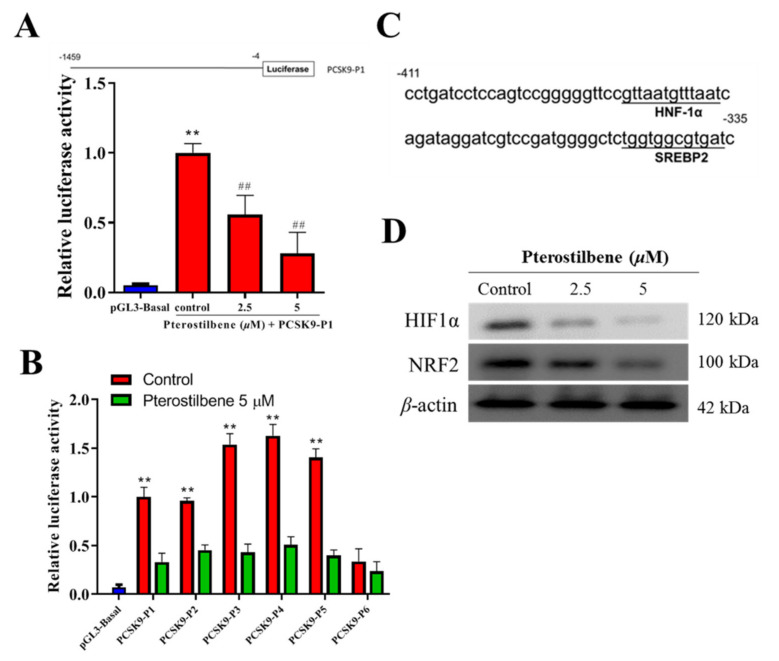
Pterostilbene suppresses the PCSK9 promoter activity and hyperlipidemia-associated transcription factors in HL-1 cells. (**A**) The luciferase reporter construct of human PCSK9 promoter. Position −4 is the 3′ end of the PCSK9 promoter inserts, common to all reporter constructs for the promoter. The 5′ ends of the promoters in each promoter-reporter construct are marked by −1459, and the name of the construct is indicated on the right (upper). Graph of the effect of pterostilbene on the luciferase activity of the PCSK9-P1 construct (lower). (**B**) Histograms showing the effect of pterostilbene on the luciferase activities of PCSK9-P1–6 constructs. (**C**) Depiction of PCSK9 promoter binding motifs for HNF1α and SREBP2 within the region from −411 bp into −335 bp. (**D**) Western blotting was performed to measure the protein expression level of HIF1α and NRF2. Representative images were from three independent experiments. ** *p* < 0.01 vs. pGL3-Basal; ## *p* < 0.01 vs. control.

**Figure 5 antioxidants-10-01280-f005:**
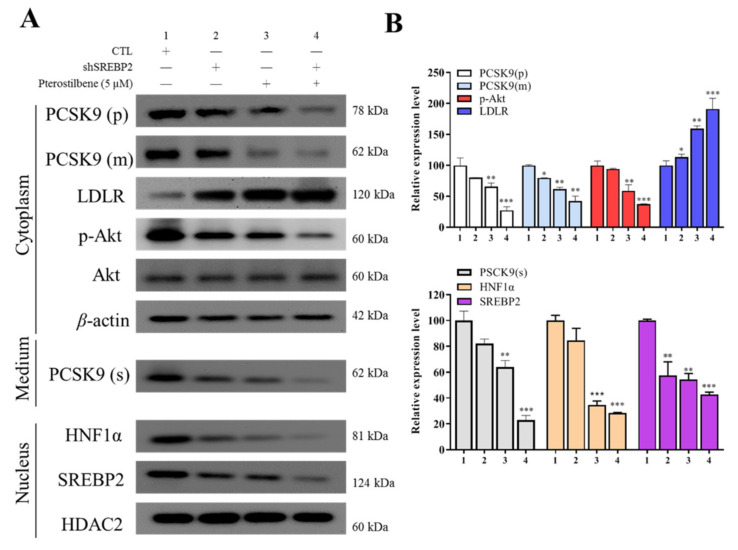
Pterostilbene as compared to short hairpin RNA against SREBP2 (shSREBP2) induces higher protein expression of LDLR and lower nuclear accumulation of HNF-1α. (**A**) Representative Western blot images of the effect of pterostilbene or shSREBP2 on cytoplasmic PCSK9(p), PCSK9(m), LDLR, p-Akt, Akt, and PCSK9(s) in medium and nuclear HNF-1α or SREBP2 proteins. β-actin and HDAC are loading controls. (**B**) Graphical representation of (**A**). * *p* < 0.05; ** *p* < 0.01; *** *p* < 0.001.

**Figure 6 antioxidants-10-01280-f006:**
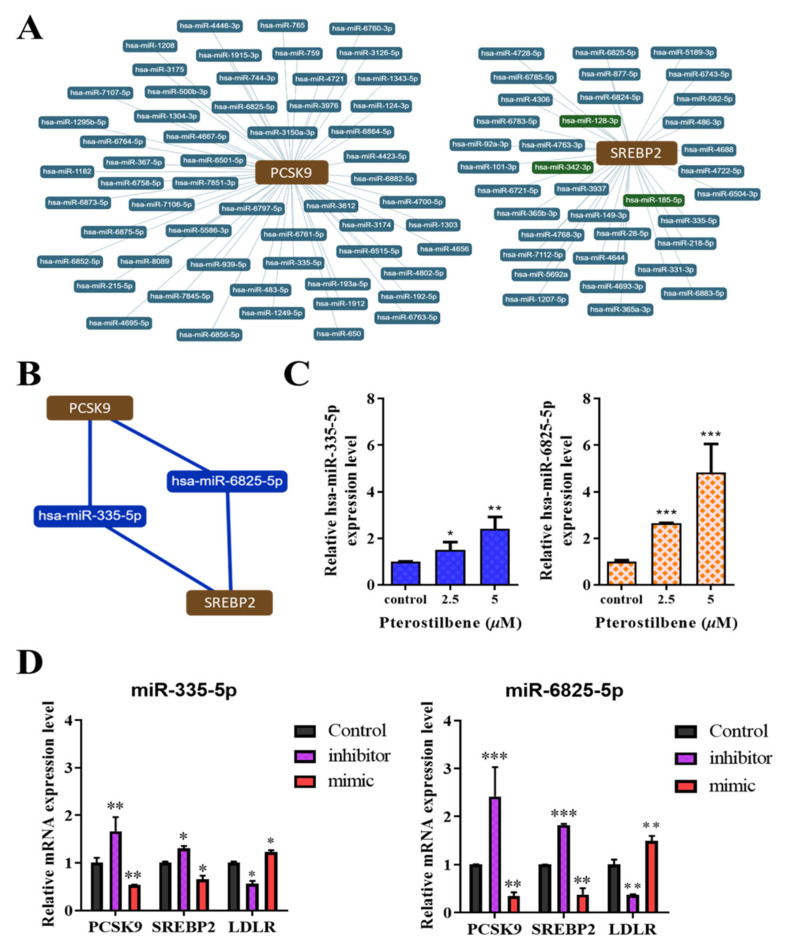
Hsa-miR-335 and has-miR-6825 mediate the PCSK9/SREBP2 interaction, and the therapeutic effect of pterostilbene involves the epigenetic regulation of PCSK9, SREBP2, and LDLR mRNA expression. (**A**) Radial connectivity chart of PCSK9 or SREBP2 and candidate miRNAs. miRNAs of interest are indicated in red outlines. (**B**) Depiction of the predicted interaction between PCSK9, hsa-miR-6825-5p, hsa-miR-335-5p, and SREBP2. (**C**) Histograms showing how pterostilbene enhances the expression of miR-335-5p *(left*) and miR-6825-5p (*right*). (**D**) Graphical representation of the effect of miR-335-5p *(left*) and miR-6825-5p (*right*) as a mimicker or inhibitor of PCSK9, SREBP2, and LDLR mRNA expression. * *p* < 0.05; ** *p* < 0.01; *** *p* < 0.001.

## Data Availability

Data is contained within the article and Appendix A.

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
