# Peer review of "Pterostilbene Increases LDL Metabolism in HL-1 Cardiomyocytes by Modulating the PCSK9/HNF1α/SREBP2/LDLR Signaling Cascade, Upregulating Epigenetic hsa-miR-335 and hsa-miR-6825, and LDL Receptor Expression"

_antioxidants, 2021, doi:10.3390/antiox10081280_

Round 1

Reviewer 1 Report

Journal: Antioxidants 

Manuscript ID: antioxidants-1295469

Title: Pterostilbene increases LDL metabolism in HL-1 cardiomyocytes by modulating the PCSK9/HNF1α/SREBP2/LDLR signaling cascade, upregulating epigenetic control of hsa-miR-335 and hsa-miR-6825, and increasing LDL receptor expression

Authors: Yen-Kuang Lin, Chi-Tai Yeh, Kuang-Tai Kuo, Vijesh Kumar Yadav, Iat-Hang Fong, Nicholas G. Kounis, Patrick Hu, Ming-Yow Hung*

Revision:

The Article entitled: "Pterostilbene increases LDL metabolism in HL-1 cardiomyocytes by modulating the PCSK9/HNF1α/SREBP2/LDLR signaling cascade, upregulating epigenetic control of hsa-miR-335 and hsa-miR-6825, and increasing LDL receptor expression" is an interesting article. The authors found positive effect of pterostilbene using  cardiomyocytes. In particular they described that cardioprotective benefits effect of pterostilbene in LDL-C lowering were mediated through signaling pathway PCSK9/HNF1α/SREBP2/LDLR involving transcription factors. This is interesting because the pterostilbene in association with other drugs (as suggest the authors) could have a possible use future clinical for benefit the patients with hyperlipidemia.

The work is clear because the objectives are well specified, the used methods are abundantly descibed and the results are described with appropriate statistical analysis and clear graphs.

I think that the manuscript is suitable for its publication in Antioxidant journal, suggesting:

  • the title needs to be reshaped because it is too long.

Author Response

We thank the reviewer for carefully reading our manuscript and providing valuable comments. We accordingly response the questions raised by the Reviewer as follows:

Point-by-point responses to reviewer’s comments:

Dear Reviewer,

Coauthors and I very much appreciated the encouraging, critical and constructive comments on this manuscript by the reviewer. We have followed the reviewer's comments thoroughly and feel that they have further helped in strengthening the manuscript.

Reviewer #1:

Comments and Suggestions for Authors

Journal: Antioxidants

Manuscript ID: antioxidants-1295469

Title: Pterostilbene increases LDL metabolism in HL-1 cardiomyocytes by modulating the PCSK9/HNF1α/SREBP2/LDLR signaling cascade, upregulating epigenetic control of hsa-miR-335 and hsa-miR-6825, and increasing LDL receptor expression

Authors: Yen-Kuang Lin, Chi-Tai Yeh, Kuang-Tai Kuo, Vijesh Kumar Yadav, Iat-Hang Fong, Nicholas G. Kounis, Patrick Hu, Ming-Yow Hung*

Revision:

The Article entitled: "Pterostilbene increases LDL metabolism in HL-1 cardiomyocytes by modulating the PCSK9/HNF1α/SREBP2/LDLR signaling cascade, upregulating epigenetic control of hsa-miR-335 and hsa-miR-6825, and increasing LDL receptor expression" is an interesting article. The authors found positive effect of pterostilbene using cardiomyocytes. In particular they described that cardioprotective benefits effect of pterostilbene in LDL-C lowering were mediated through signaling pathway PCSK9/HNF1α/SREBP2/LDLR involving transcription factors. This is interesting because the pterostilbene in association with other drugs (as suggest the authors) could have a possible use future clinical for benefit the patients with hyperlipidemia.

The work is clear because the objectives are well specified, the used methods are abundantly descibed and the results are described with appropriate statistical analysis and clear graphs.

I think that the manuscript is suitable for its publication in Antioxidant journal, suggesting: the title needs to be reshaped because it is too long.

A: We are grateful to the Reviewer and feel overwhelmed to the reviewer’s encouraging comment on our manuscript. As per the suggestions we have tried to trim the title and make it short. Please kindly see our revised title in the main text of the manuscript.

Title: Pterostilbene increased LDL metabolism in HL-1 cardiomyocytes through PCSK9/HNF1α/SREBP2/LDLR signaling cascade, upregulating epigenetic hsa-miR-335 and hsa-miR-6825, and LDL receptor expression

Reviewer 2 Report

In the present manuscript, the authors have been investigated the influence of the polyphenolic compound pterostilbene on LDL metabolism in HL-1 cardiomyocytes. For this purpose, the authors have employed the following analyses: Western blotting, flow cytometry, immunofluorescence staining, and mean fluorescence intensity analyses. The authors have shown that pterostilbene dose-dependently decreases and increases the protein and mRNA expression of PCSK9 and LDLR, respectively. The cardioprotective effects of pterostilbene in LDL-C lowering were mediated through PCSK9/HNF1α/SREBP2/LDLR signalling pathway. The mechanisms of action involve inhibiting HNF1α/SREBP2/HIF1α/Nrf2 transcription factors, decreasing the secretion of PCSK9, up-regulating epigenetic control of miR-335 and miR-6825, and increasing LDLR expression. The novel inhibitory pathways of pterostilbene, related to transcriptional and epigenetic PCSK9 antagonism, and LDLR upregulation, are recognized as possible lead to new approaches to cardiovascular prevention.

The whole manuscript is very well written. The results are very well presented. All Figures are informative and professionally prepared. The abstract and conclusion are concise and punctual.

In my opinion, the present manuscript can be published in the Antioxidants after minor revision.

Authors should consider adding the following reference:

Involvement of Antioxidant Defenses and NF-κB/ERK Signaling in Anti-inflammatory Effects of Pterostilbene, a Natural 3 Analogue of Resveratrol, Appl. Sci. 202111(10), 4666.

Author Response

Reviewer #2:

Comments and Suggestions for Authors

In the present manuscript, the authors have been investigated the influence of the polyphenolic compound pterostilbene on LDL metabolism in HL-1 cardiomyocytes. For this purpose, the authors have employed the following analyses: Western blotting, flow cytometry, immunofluorescence staining, and mean fluorescence intensity analyses. The authors have shown that pterostilbene dose-dependently decreases and increases the protein and mRNA expression of PCSK9 and LDLR, respectively. The cardioprotective effects of pterostilbene in LDL-C lowering were mediated through PCSK9/HNF1α/SREBP2/LDLR signalling pathway. The mechanisms of action involve inhibiting HNF1α/SREBP2/HIF1α/Nrf2 transcription factors, decreasing the secretion of PCSK9, up-regulating epigenetic control of miR-335 and miR-6825, and increasing LDLR expression. The novel inhibitory pathways of pterostilbene, related to transcriptional and epigenetic PCSK9 antagonism, and LDLR upregulation, are recognized as possible lead to new approaches to cardiovascular prevention.

The whole manuscript is very well written. The results are very well presented. All Figures are informative and professionally prepared. The abstract and conclusion are concise and punctual.

In my opinion, the present manuscript can be published in the Antioxidants after minor revision.

Authors should consider adding the following reference:

Involvement of Antioxidant Defenses and NF-κB/ERK Signaling in Anti-inflammatory Effects of Pterostilbene, a Natural 3 Analogue of Resveratrol, Appl. Sci. 2021, 11(10), 4666.

A: We are grateful for the Reviewer appreciation for our manuscript and feel overwhelmed, as per the suggestions we have cited the aforementioned article in our manuscript main text. Please kindly see our revised text discussion section in line 472-474.

Please also see our updated reference sections in line 65-658.

  1. Jayakumar, T.; Wu, M.-P.; Sheu, J.-R.; Hsia, C.-W.; Bhavan, P.S.; Manubolu, M.; Chung, C.-L.; Hsia, C.-H. Involvement of an-tioxidant defenses and nf-κb/erk signaling in anti-inflammatory effects of pterostilbene, a natural analogue of resveratrol. Appl. Sci. 2021, 11, 4666.
